



# Assessment of extreme flows and uncertainty under climate change: disentangling the contribution of RCPs, GCMs and internal climate variability

Chao Gao[1,2], Martijn J. Booij[2], Yue-Ping Xu[1]

[1]Institute of Hydrology and Water Resources, Zhejiang University, Hangzhou 310058, China
[2] Water Engineering and Management Group, Faculty of Engineering Technology, University of Twente, Enschede 7500 AE, The Netherlands

*Correspondence*: Yue-Ping Xu (yuepingxu@zju.edu.cn)

**Abstract.** Projections of streamflow, particularly of extreme flows under climate change are essential for future water
resources management and development of adaptation strategies to floods and droughts. However, these projections are subject
to uncertainties originating from different sources. In this study, we explore the possible changes in future streamflow,
particularly for high and low flows, under climate change in the Qu River basin, East China. ANOVA (Analysis of Variance)
is employed to quantify the contribution of different uncertainty sources from RCPs (Representative Concentration Pathways),
GCMs (Global Climate Models) and internal climate variability, using an ensemble of four RCP scenarios, nine GCMs and
1,000 simulated realizations of each model-scenario combination by SDRM-MCREM (a stochastic daily rainfall model
coupling a Markov chain model with a rainfall event model). The results show that annual mean flow and high flows are
projected to increase and low flows will probably decrease in 2041-2070 (2050s) and 2071-2100 (2080s) relative to the
historical period 1971-2000, suggesting a higher risk of floods and droughts in the future in the Qu River basin, especially for
the late 21st century. Uncertainty in mean flows is mostly attributed to GCM uncertainty. For high flows and low flows,
internal climate variability and GCM uncertainty are two major uncertainty sources for the 2050s and 2080s, while for the
2080s, the effect of RCP uncertainty is becoming more pronounced, particularly for low flows. The findings in this study can
help water managers to get a better knowledge and understanding of streamflow projections and support decision making on
adaptions to changing climate under uncertainty in the Qu River basin.

## 1 Introduction

Climate change has been demonstrated to produce profound impacts on hydrological processes all over the world, with its
effects lasting throughout the whole 21st century (Bosshard et al., 2013;Addor et al., 2014). Future streamflow projections
offer a valuable basis for the assessment of various hydrological extremes including floods and droughts (Giuntoli et al., 2018),
which is beneficial for decision makers to plan effective countermeasures for a changing climate (Addor et al., 2014). However,
these climate change projections are usually subject to high uncertainty, making it difficult to identify robust adaptation



strategies in the decision process (Whateley and Brown, 2016). Therefore, it is of fundamental importance to characterize and quantify uncertainty associated with projections in climate change impact studies (Deser et al., 2010).

Uncertainty in climate change projections mainly arises from three different sources, i.e. scenario uncertainty, model uncertainty and internal climate variability (Evin et al., 2019;Deser et al., 2010). Scenario uncertainty is interpreted as responses to different assumptions of future greenhouse gas emissions, which reflects the limited knowledge of external factors

such as anthropogenic activities and social development strategies, that influence the climate system (Nakicenovic and Swart, 2000). Model uncertainty originates from different responses of different model structures under the same future emission scenario that are mainly due to imperfect physical and numerical formulations representing the actual climate system (Stocker et al., 2014). Internal climate variability is the natural unforced variability of the climate system representing dynamical processes intrinsic to the ocean, the atmosphere and the coupled system (Deser et al., 2010). The former two sources of

uncertainty are usually estimated using a multi-model ensemble of climate projections derived from different representative concentration pathways (RCPs) and a large number of regional climate models (RCMs) or global climate models (GCMs), respectively. It is considered possible to potentially reduce the uncertainty of these two sources by improving our scientific knowledge in accurately predicting future emissions and interpreting geophysical processes (Lafaysse et al., 2014). However, the internal climate variability can not be reduced and will persistently exist because of its inherent property (Fatichi et al.,

2016), and it is typically evaluated with members or runs representing different initial conditions for the same climate model under the same emission scenario.

These different uncertainty sources in climate projections have been quantified by multiple studies (Yip et al., 2011;Zhuan et al., 2018;Evin et al., 2019). The relative importance varies depending on factors like the type of climate variable and temporal and spatial scales (Zhuan et al., 2018). For example, many studies have demonstrated that model uncertainty is generally

dominant in rainfall projections rather than scenario uncertainty throughout the 21st century, while scenario uncertainty becomes gradually more important in the late 21st century, particularly for temperature projections (Zhuan et al., 2018;Yip et al., 2011). In the near future, internal climate variability contributes largely to the total uncertainty, especially for rainfall projections, and becomes more important with decreasing temporal and spatial scales (Hingray and Saïd, 2014;Giorgi, 2002). A question arising is how these uncertainty sources in climate projections will affect future streamflow projections? In recent

years, different sources of uncertainty in streamflow projections have also been investigated (Bosshard et al., 2013;De Niel et al., 2019). Vetter et al. (2016) assessed different uncertainty sources in projections of hydrological changes using four RCPs, five GCMs and nine hydrological models (HMs), and concluded that GCMs generally resulted in the largest uncertainty contribution, followed by RCPs and HMs. De Niel et al. (2019) adopted a large multi-model ensemble consisting of different RCPs, GCMs, downscaling methods, hydrological model structures and hydrological parameter sets to comprehensively

evaluate the uncertainty existing in peak flows. The results showed that compared to the dominant uncertainty derived from GCMs and RCPs, HMs and parameter sets are less important for peak flows. Previous studies indicated that uncertainty originating from climate projections is generally larger than uncertainty in the hydrological simulation process (Teng et al., 2012;Karlsson et al., 2016). However, the majority of the studies only considered the uncertainty caused by scenarios and



climate models, but neglected the effect of internal climate variability on streamflow projections, although Jung et al. (2011)

and Kay et al. (2008) demonstrated that natural variability is also critical with respect to hydrological changes. The focus of our study is quantifying how uncertainty of climate projections, i.e. scenario uncertainty, model uncertainty and internal climate variability, is propagated into streamflow projections, without taking into account uncertainty embedded in the simulation of hydrological processes.

Recently, analysis of variance (ANOVA) has been a widely-used approach to quantify the contributions of different

uncertainty sources in climate change impact studies (Qi et al., 2016;Vetter et al., 2014;Bosshard et al., 2013). ANOVA is a model-based method to partition the total variance into different contributing components of variation. The advantage of ANOVA is that it can not only interpret main single factors, but interactions of factors as well (Vetter et al., 2014). However, there is a problem in quantifying the contribution of internal climate variability using ANOVA since multiple members for each model are required (Evin et al., 2019). This multiple-member constraint results in many single-member models being

discarded (Bracegirdle et al., 2014). Thus, only a limited number of climate models can be selected and the climate projection information cannot be fully utilized. To solve this problem, a simple alternative way is to use weather generators or stochastic rainfall models to generate multiple members for each model-scenario combination to interpret its internal climate variability, similarly to Lafaysse et al. (2014) and Fatichi et al. (2016). When using these methods, the accuracy of weather generators and stochastic rainfall models is very essential. In this study, we adopt a newly developed stochastic rainfall model by the authors,

named SDRM-MCREM (a stochastic daily rainfall model coupling a Markov chain model with a rainfall event model) (Gao et al., 2020a). Compared to previous stochastic rainfall models, SDRM-MCREM can comprehensively preserve rainfall characteristics of both rainfall time-series (e.g. monthly mean rainfall and various rainfall percentiles) and rainfall event characteristics (e.g. different classes of rainfall duration, rainfall depth, and dry spell events and rainfall temporal patterns of different rainfall types). These are all important for streamflow generation. Furthermore, outputs of SDRM-MCREM can well

be used as inputs into a hydrological model to reproduce streamflow extremes to effectively conduct flood and drought risk assessment. Similarly, this study also employs ANOVA to quantify the contribution of different uncertainty sources.

The contribution of this study is that we adopt a well-performing stochastic rainfall model to better reflect internal climate variability, and then robustly investigate changes in streamflow projections and uncertainty therein propagated from different sources of climate uncertainty under climate change. We use four RCPs and nine GCMs to account for the uncertainty of

scenarios and climate models, respectively. In this study, our main targets are: (1) to look into the overall change of streamflow, particularly for high flows and low flows, in the mid-future period (2041-2070) and the far-future period (2071-2100); (2) to quantify the contribution rate of different uncertainty sources, i.e. RCP uncertainty, GCM uncertainty and internal climate variability, in streamflow projections; and (3) to get insight into how different sources of uncertainty evolve with time in the future. Our study contributes to a better understanding of changes of hydrological extremes and provides useful information

for designing adaptation strategies to flood and drought events under a changing climate.





## 2 Study area and data

The Qu River basin (Fig. 1) is used as the study area, that is situated in the western part of Zhejiang Province, East China. The Qu River basin has an area of 5,536 km2, covering longitudes from 118° to 119°E and latitudes from 28° to 29°30′N, and is characterized by the Asian subtropical monsoon climate with a hot rainy summer and cold dry winter (Gao et al., 2020c). The

annual mean temperature is around 15~18℃ and the average annual rainfall is around 1800 mm with more than 50% occurring in April to July (Gao et al., 2020b). Available data were from 14 gauged rainfall stations, three meteorological stations and one hydrological station with observed rainfall, temperature and streamflow data for the historical period 1970-2000.

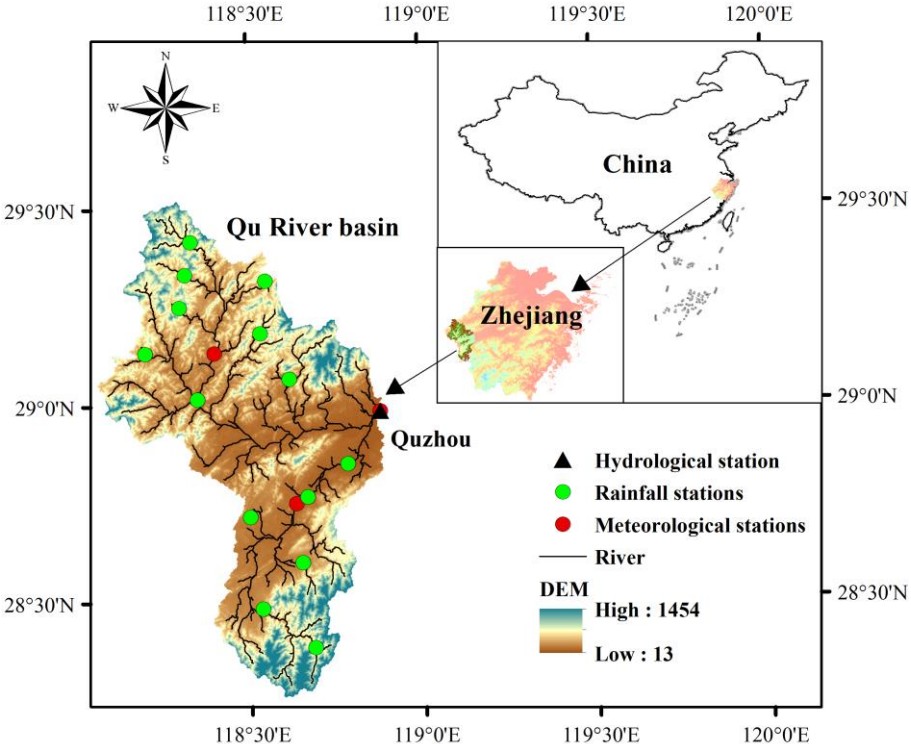

**Figure 1:** Study area and distribution of rainfall, meteorological and hydrological stations.

To conduct climate change impact analysis, projections of daily rainfall, daily maximum temperature, daily minimum temperature and daily mean temperature were obtained from nine GCMs (Table 1) for the mid-future period 2041-2070 (i.e. 2050s) and the far-future period 2071-2100 (i.e. 2080s) under four RCP emission scenarios, including RCP8.5, RCP6.0, RCP4.5, and RCP2.6. The historical period is 1971-2000, consistent with the observed data. The nine GCMs had been selected out of 17 GCMs based on the evaluation results of Gao et al. (2020b), which showed that these GCMs can simultaneously well

preserve rainfall time-series statistics and rainfall event characteristics after being bias corrected compared to other GCMs. The information of the nine selected GCMs is listed in Table 1.





**Table 1.** Name, Research Institution and Resolution of the nine selected GCMs

| Name of GCMs | Research Institution, Country | Lon ×Lat Resolution |
|---|---|---|
| NorESM1-M | Norwegian Climate Center, Norway | ~2.5000°×1.8947° |
| MIROC-ESM-CHEM | Center for Climate System Research, Japan | ~2.8125°×2.7906° |
| MIROC-ESM | | ~2.8125°×2.7906° |
| IPSL-CM5A-MR | Institute Pierre-Simon Laplace, France | ~2.5000°×1.2676° |
| IPSL-CM5A-LR | | ~3.7500°×1.8947° |
| HadGEM2-ES | Met Office Hadley Centre, UK | ~1.8750°×1.2500° |
| HadGEM2-AO | | |
| GFDL-ESM2G | Geophysical Fluid Dynamics Laboratory, America | ~2.5000°×2.0225° |
| BCC-CSM1-1 | Beijing Climate Center, China | ~2.8125°×2.7673° |

## 3 Methodology

The research was carried out at the catchment scale. Thiessen polygon method (Thiessen, 1911) was used to calculate the areal-averaged daily rainfall and daily maximum, minimum and mean temperatures in the Qu River basin. The areal-averaged daily potential evapotranspiration (PET) was computed using the areal-averaged daily maximum and minimum temperatures and the Hargreaves method (Hargreaves and Zohrab, 1985).

### 3.1 Bias correction method for rainfall and temperature

Since current GCMs have insufficient spatial resolutions for the catchment scale, GCM outputs often show large deviations compared to observed data and cannot be directly applied for impact studies (Teng et al., 2015;Räty et al., 2014). Therefore, many bias correction approaches have been developed to correct the bias between GCM outputs and observations (Teutschbein and Seibert, 2012). In this study, we used the distribution mapping (DM) method to correct GCM-simulated climate variables. The underlying idea is to identify the possible bias between GCM simulations and observations by projecting the distribution functions of GCM-simulated climate variables onto those of observations in the historical period (Eq. (1)) and then use the determined bias as a basis to correct future GCM simulations (Eq. (2)) (Miao et al., 2016).

$$\tilde{x}_{sim,his} = F_{obs}^{-1}\left( F_{sim,his}\left( x_{sim,his} \right) \right) \tag{1}$$

$$\tilde{x}_{sim,fut} = x_{sim,his} + F_{obs}^{-1}\left( F_{sim,fut}\left( x_{sim,fut} \right) \right) - F_{sim,his}^{-1}\left( F_{sim,fut}\left( x_{sim,fut} \right) \right) \tag{2}$$

in which, $x_{sim,his}$ and $x_{sim,fut}$ are the raw GCM simulations in the historical and future periods; $F_{sim,his}$ and $F_{sim,fut}$ are their corresponding cumulative distribution functions (CDFs); $\tilde{x}_{sim,his}$ and $\tilde{x}_{sim,fut}$ are the corresponding historical and future bias-





corrected GCM simulations; $F_{sim,his}^{-1}$ and $F_{obs}^{-1}$ are the inverse CDFs of GCM simulations and observations in the historical period.

A double-gamma distribution that consists of two single gamma distributions was adopted in this study to fit rainfall (Räty et al., 2014). Firstly, GCM-simulated and observed rainfall are respectively divided into two parts by their 95th percentiles, i.e. one part represents the normal rainfall (smaller than the 95th percentile) and the rest part represents the extreme rainfall (equal to or larger than the 95th percentile), and each part is subsequently fitted and bias corrected using the single gamma distribution shown in Eq. (3). The advantage of the double gamma distribution is that he features of extreme rainfall

and normal rainfall can be captured at the same time (Olsson et al., 2015). The detailed bias correction procedure and the solution of the problem of excessive drizzle days existing in GCM-simulated rainfall can be found in Gao et al. (2020b).

For temperature series, the Gaussian distribution shown in Eq. (4) was used (Teutschbein and Seibert, 2012). The historical and future GCM-simulated temperatures were bias corrected according to Eq. (1) and Eq. (2), respectively. The method shown in Eq. (2) explicitly considers the changes between the historical and future distributions of GCM-simulated

temperatures, which remains the same change factors (i.e. the difference between the future and historical monthly mean temperatures) before and after bias correction. For GCM-simulated rainfall and temperature series, the bias correction processes were conducted at the monthly scale.

$$f_1\left(x_1 | \alpha_1, \beta_1\right) = \frac{\beta_1^{\alpha_1} \cdot e^{-\beta_1 x} \cdot x^{\alpha_1 - 1}}{\Gamma\left(\alpha_1\right)} \,, \, f_2\left(x_2 | \alpha_2, \beta_2\right) = \frac{\beta_2^{\alpha_2} \cdot e^{-\beta_2 x} \cdot x^{\alpha_2 - 1}}{\Gamma\left(\alpha_2\right)} \tag{3}$$

$$f\left(x | \mu, \sigma^2\right) = \frac{1}{\sigma \cdot \sqrt{2\pi}} e^{\frac{-(x-\mu)^2}{2\sigma^2}} \tag{4}$$

in which, $\alpha_1, \beta_1$ and $\alpha_2, \beta_2$ are the parameters of two single-gamma distributions, and $\mu, \sigma$ are the parameters of the Gaussian distribution.

### 3.2 Stochastic daily rainfall model

The new stochastic daily rainfall model coupling a Markov chain model with a rainfall event model (SDRM-MCREM) was developed by Gao et al. (2020a). The framework of SDRM-MCREM is shown in Fig.2 and consists of the following steps:

(1)  The time series of rainfall occurrence with values of 1 (wet day) and 0 (dry day) is generated using a third-order Markov chain model. A wet day is defined when daily rainfall is larger than or equal to 0.1 mm, otherwise it is a dry day.

(2)  Based on the occurrence of wet and dry days, rainfall events are extracted and the rainfall duration of each rainfall event is subsequently determined. In this study area, when continuous wet days are separated by one or more

160        continuous dry days, the continuous wet days are considered as a rainfall event.





(3)   Given rainfall durations in Step 2, the rainfall depth (i.e. the total amount) of each rainfall event is simulated using a fitted conditional copula function. Copula functions are used to construct the joint probability distribution of the dependent rainfall depth and duration (Gao et al., 2018). The best-fitted probability distributions of rainfall duration and depth and the best-fitted copula function are selected based on the Akaike information criterion (AIC).

(4)   The event class of each rainfall event is determined according to its depth and duration. Rainfall depths are classified into light, moderate, heavy and extreme events, and rainfall durations are classified into short, medium, long and extreme events, just like the table shown in Fig. 2.

(5)   Following Step 4, one rainfall type is generated for each rainfall event based on the occurrence probability of different rainfall types for the given rainfall event class, and the corresponding dimensionless temporal pattern of this specific
rainfall type is stochastically simulated. According to the location of a rainfall peak within one rainfall event, i.e. early, middle or late stage, the dimensionless temporal patterns of gauged rainfall events are grouped into three different rainfall types, namely Delayed-type (D), Central-peaked-type (C) and Advanced-type (A).

(6)   For each rainfall event, its depth and duration are temporally allocated according to the dimensionless rainfall pattern to form the complete rainfall event. After allocating all the rainfall events, the complete rainfall time series can be
obtained.

Detailed information about the adopted probability distributions of rainfall depth and duration, copula functions, classification of rainfall event classes, rainfall types and the simulation procedure of temporal patterns of different rainfall types (step 3-5) in SDRM-MCREM can be found in Gao et al. (2020a). Using the constructed SDRM-MCREM, nine GCM bias-corrected rainfall time series for the historical period of 30 years and the two future periods of 30 years under four RCPs
were stochastically simulated 1,000 times.





**Figure2:** Framework of SDRM-MCREM.

## 3.3 Hydrological model

The parsimonious and effective conceptual rainfall-runoff model with four parameters GR4J (Perrin et al., 2003) was used for

hydrological modelling. The GR4J model has shown good performances in different climate regions (e.g. Van Esse et al. (2013) and Tian et al. (2013)), including the Asian subtropical monsoon climate region (Gao et al., 2020c;Tian et al., 2015). Thus, this model can be applied in the Qu River basin. The required input data are the drainage area, areal-averaged daily rainfall and areal-averaged daily PET, and the output is daily streamflow data. In this study, rigorous calibration and validation were carried out for GR4J: (1) a large number of parameter sets, i.e. 30,000, is generated using Latin hypercube sampling to calibrate



the model; (2) the split-sample cross validation method (Gao et al., 2020b;Chen et al., 2008) is employed to comprehensively

select the optimum parameter set. The observed data of 1970 are used as the warming-up period, the first 20 years (1971-1990)

are used for calibration and the last 10 years (1991-2000) for validation. Conversely, the last 20 years 1981-2000 are used for

calibration and the first 10 years 1971-1980 for validation. The comprehensive evaluation indicator Y that combines NS (Nash-

Sutcliffe coefficient) and RVE (Relative volume error) (Akhtar et al., 2009) in Eq. (5) ~ Eq. (7) is adopted as the objective

function. When the average of Y in the two calibration periods reaches its highest value, the parameter set is considered to be

the optimum one.

$$Y = \frac{NS}{1 + |RVE|} \tag{5}$$

$$NS = 1 - \frac{\sum_{i=1}^{N} \left[ Q_m(i) - Q_o(i) \right]^2}{\sum_{i=1}^{N} \left[ Q_o(i) - \overline{Q_o} \right]^2} \tag{6}$$

$$RVE = 100 \cdot \frac{\sum_{i=1}^{N} \left[ Q_m(i) - Q_o(i) \right]}{\sum_{i=1}^{N} Q_o(i)} \tag{7}$$

in which, $N$ is the length of the time series, $Q_m(i)$ and $Q_o(i)$ are the simulated and observed streamflow of the $i$th day, $\overline{Q_o}$

is the mean observed streamflow.

The GR4J model with the optimal parameter set was used to generate 1,000 realizations of 30-year streamflow data for

each GCM in the historical period and the two future periods under four RCPs, driven by the corresponding 1,000 realizations

of rainfall time series simulated by SDRM-MCREM and the PET data calculated using the Hargreaves method for each GCM.

It should be noted that there is only one set of PET data for each GCM as its daily temperature data is not stochastically

simulated.

**3.4 Contribution of different uncertainty sources**

In this study, ANOVA was adopted to calculate the contribution of different uncertainty sources, which is an effective tool

than can decompose the total variance into variances of different sources and thus quantify the proportion of variance of

different sources in the total variance. ANOVA has been widely used in climate change uncertainty analysis in recent years

(Vetter et al., 2014;Qi et al., 2016;Vetter et al., 2016). As the 1,000 realizations of each GCM in the historical and future

periods are all stochastically simulated from the same GCM under a specific RCP scenario (i.e. model-scenario combination),

the stochastic uncertainty (i.e. internal climate variability) is considered as a set of independent realizations from an infinite

population, known as the within-group variation. Under this condition, there are two main effect factors (known as the between-

group variation), i.e. GCMs and RCPs. Therefore, the two-way ANOVA analysis (Yip et al., 2011) was employed for this

study.



The total sum of variance ($SST$) can be split into four parts: stochastic error square sum ($SS_{Stoc}$), main effect of RCPs ($SS_{RCP}$) and GCMs ($SS_{GCM}$), and interaction effect of RCPs and GCMs ($SS_{RCP \times GCM}$) (Northrop and Chandler, 2014), which are described in Eq. (8) ~ Eq. (14). To explicitly compare the relative importance of internal climate variability, GCM uncertainty and RCP uncertainty, the interaction term of RCPs and GCMs is firstly divided by equal shares and then added to the respective factors of RCPs and GCMs. The contribution of the three uncertainty sources is calculated according to Eq. (14).

$$SST = \sum_{i=1}^{N_{rcp}} \sum_{j=1}^{N_{gcm}} \sum_{k=1}^{N_{stoc}} \left( y_{ijk} - \bar{y} \right)^2 \tag{8}$$

$$SST = SS_{Stoc} + SS_{RCP} + SS_{GCM} + SS_{RCP \times GCM} \tag{9}$$

$$SS_{Stoc} = \sum_{i=1}^{N_{rcp}} \sum_{j=1}^{N_{gcm}} \sum_{k=1}^{N_{stoc}} \left( y_{ijk} - \bar{y}_{ij\bullet} \right)^2 \tag{10}$$

$$SS_{RCP} = N_{gcm} N_{stoc} \sum_{i=1}^{N_{rcp}} \left( \bar{y}_{i\bullet\bullet} - \bar{y} \right)^2 \tag{11}$$

$$SS_{GCM} = N_{rcp} N_{stoc} \sum_{j=1}^{N_{gcm}} \left( \bar{y}_{\bullet j\bullet} - \bar{y} \right)^2 \tag{12}$$

$$SS_{RCP \times GCM} = N_{stoc} \sum_{i=1}^{N_{rcp}} \sum_{j=1}^{N_{gcm}} \left( \bar{y}_{ij\bullet} - \bar{y}_{i\bullet\bullet} - \bar{y}_{\bullet j\bullet} + \bar{y} \right)^2 \tag{13}$$

$$\eta^2_{Stoc} = \frac{SS_{Stoc}}{SST}, \quad \eta^2_{RCP} = \frac{SS_{RCP}}{SST} + \frac{1}{2} \frac{SS_{RCP \times GCM}}{SST}, \quad \eta^2_{GCM} = \frac{SS_{GCM}}{SST} + \frac{1}{2} \frac{SS_{RCP \times GCM}}{SST} \tag{14}$$

in which, $N_{rcp}$, $N_{gcm}$ and $N_{stoc}$ are the number of RCPs, GCMs and stochastic simulations, respectively; $y_{ijk}$ is the particular value corresponding to the $k$th simulation of the $j$th GCM under the $i$th RCP scenario; $\bar{y}$ is the overall mean; $y_{i\bullet\bullet}$, $y_{\bullet j\bullet}$ and $y_{ij\bullet}$ are the mean of all values under the particular index, respectively; $\eta^2_{Stoc}$, $\eta^2_{GCM}$ and $\eta^2_{RCP}$ are the contribution of internal climate variability, GCM uncertainty and RCP uncertainty, respectively.

## 4 Results

### 4.1 Bias correction of rainfall and temperature

The empirical cumulative distribution functions (ECDF) and monthly means of simulated climate variables from nine GCMs before and after bias correction using the DM method are compared to those of observations in the historical period 1971-2000 (shown in Fig.3). It can be clearly seen that before bias correction, the ability of GCMs to reproduce rainfall and temperature values is relatively poor. In this study area, the majority of the GCMs tends to underestimate rainfall as well as temperature. All rainfall and temperature bias-corrections significantly improve the raw GCM simulations, and currently the bias-corrected GCM simulations are very close to the observations, which is indicated by the better matching ECDFs and monthly means





between observations and bias-corrected GCM simulations. In addition, the monthly-scale bias correction method can effectively remove the mismatch of annual temporal pattern of climate variables between observations and GCM simulations, particularly for rainfall. The detailed evaluation results of the bias-corrected rainfall of the nine GCMs can be found in Gao et al. (2020b). The results further demonstrated that the GCM simulated rainfall after bias correction can well preserve rainfall

time-series characteristics like standard deviation, coefficient of variation and various rainfall quantiles, and simultaneously well reproduce rainfall event characteristics including different rainfall event classes and rainfall patterns. Since the bias-corrected temperature data usually performed better than rainfall (Teutschbein and Seibert, 2012), the detailed evaluation results of bias-corrected temperature are not given here. Subsequently, the future GCM simulated rainfall and temperature data under four RCPs for two future periods 2050s and 2080s were bias corrected according to Eq. (2).

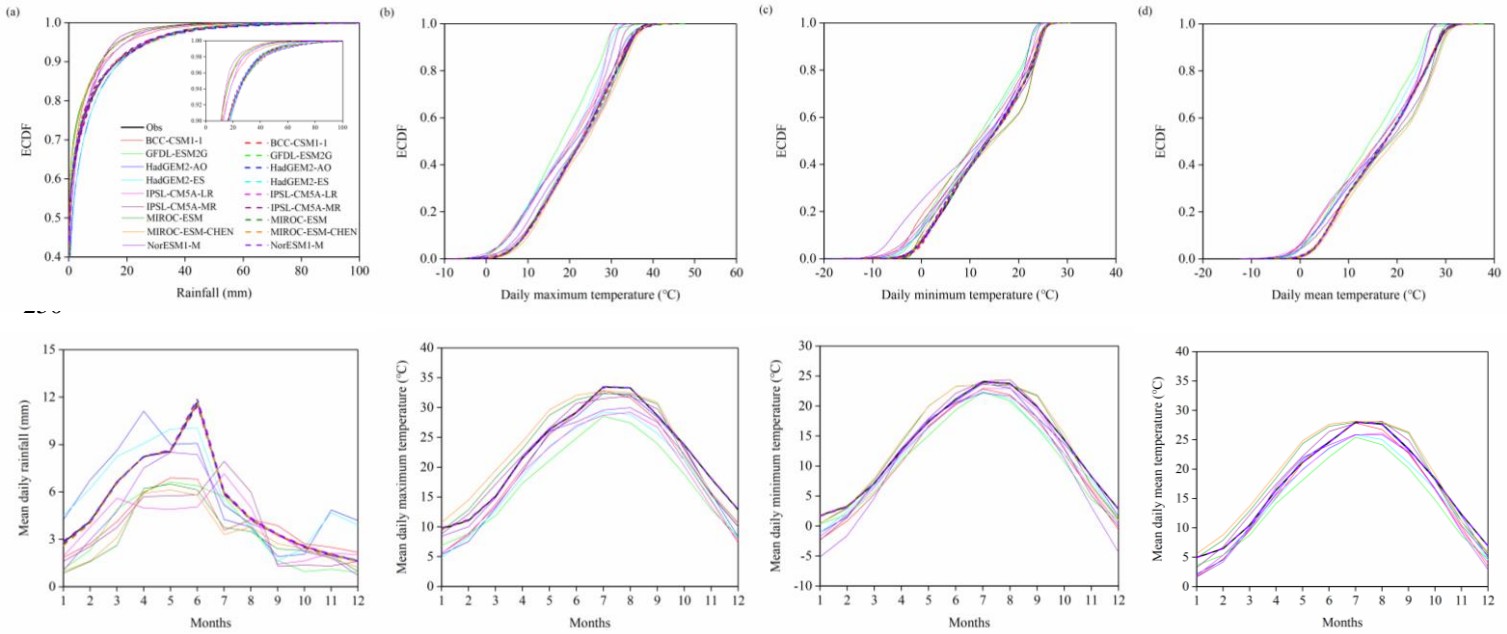

**Figure 3:** Comparison of GCM simulations before and after bias correction with observations in the empirical cumulative distributions and monthly means of (a) daily rainfall, (b) daily maximum temperature, (c) daily minimum temperature and (d) daily mean temperature. The black line represents the observed values. The colorful lines represent raw GCM simulated values. The colorful dotted lines represent bias-

corrected GCM simulated values.

The ensemble averages of the nine GCM bias-corrected monthly mean rainfall and daily maximum, minimum and mean temperatures for the historical period and the two future periods are compared in Fig. 4. The results show that daily maximum, minimum and mean temperatures all present a consistent and stable increasing trend in the future periods being the largest for RCP8.5 followed by RCP6.0, RCP4.5 and RCP2.6, and always with larger increases for the 2080s compared to the 2050s. On

average, daily maximum, mean and minimum temperatures for each month rise by 2.44 ℃, 2.31 ℃ and 2.19 ℃ in the 2050s and by 3.46 ℃, 3.29 ℃ and 3.12 ℃ in the 2080s, respectively. For rainfall, the change is not as distinct and regular as for




temperature and different RCP scenarios show different changes in different seasons. Rainfall in the monsoon season (April-June) and September generally increases in the 2050s and 2080s with larger increases in the 2080s, and no obvious changes occur in other months. In summary, the annual rainfall in the 2050s and 2080s is expected to rise by 3.18% and 5.20%, respectively.

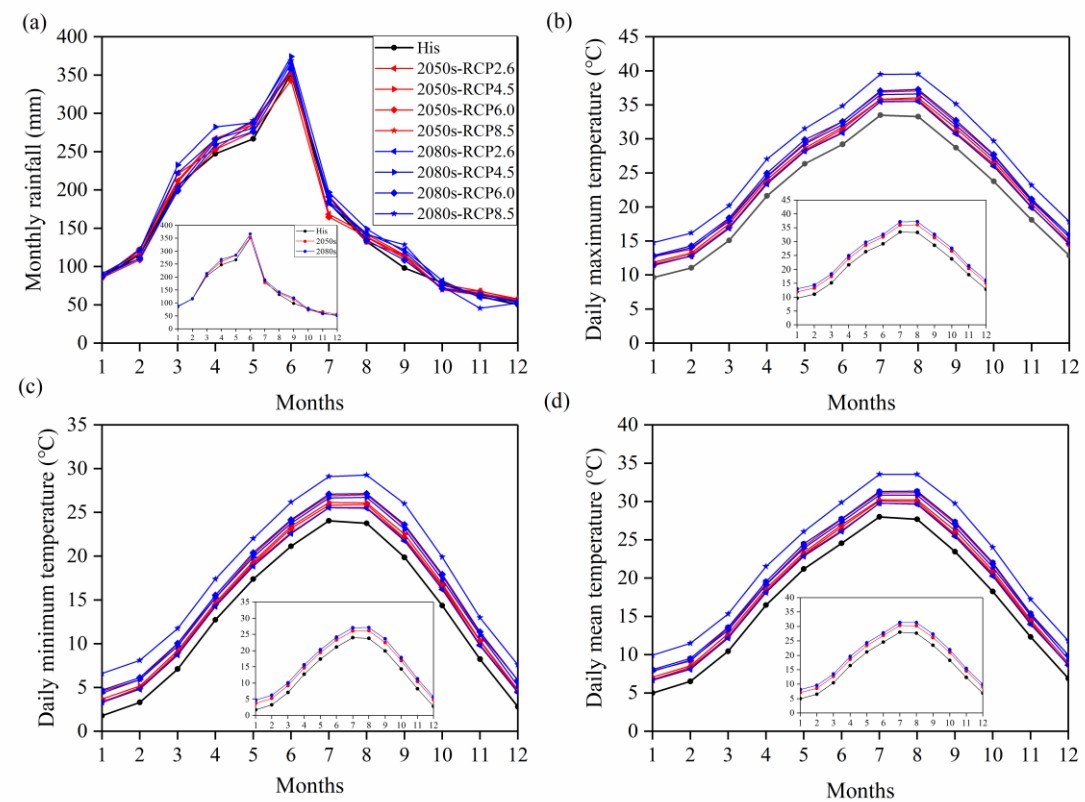

**Figure 4:** Ensemble average of GCM simulated monthly mean (a) rainfall, (b) daily maximum temperature, (c) daily minimum temperature and (d) daily mean temperature for the historical and future periods under four RCPs after bias correction.

## 4.2 Performances of stochastic rainfall model and hydrological model

### 4.2.1 Evaluation of SDRM-MCREM

The stochastic rainfall model SDRM-MCREM was evaluated for two aspects: rainfall time-series characteristics and rainfall event characteristics, through comparison of observations and the range of 1,000 simulations based on the observed rainfall in the historical period 1971-2000. The performance of SDRM-MCREM is categorized into three classes, i.e. good, fair and poor, when the observations are located in the 10th-90th percentile range, the minimum-maximum range and outside of the range of 1,000 simulations, respectively. The performance of SDRM-MCREM for the rainfall time-series characteristics, i.e. monthly mean rainfall and cumulative probability distributions including various rainfall percentiles, is shown in Fig. 5. The results





show that SDRM-MCREM performs very well for most of these rainfall characteristics, but the larger rainfall percentiles (90th-95th) are slightly underestimated. The exceedance probability distributions of rainfall extremes, including annual maximum 1-, 3- and 5-day rainfall are also assessed and presented in Fig. 6. Except that the annual maximum 1-day rainfall is

slightly overestimated by SDRM-MCREM with a fair performance, the annual maximum 3- and 5-day rainfall are all reproduced very well, as observations and simulated medians are very close. For rainfall event characteristics, detailed evaluation of SDRM-MCREM can be found in Gao et al. (2020a), including the distribution of wet and dry spells, the occurrence frequency of different classes of rainfall duration, depth and dry spell events, the temporal patterns of different rainfall types and even the occurrence frequency of different rainfall types in different classes of rainfall events. The results

demonstrated that SDRM-MCREM presents a good performance for these rainfall event characteristics as well.

In general, SDRM-MCREM can well reproduce most of the rainfall time-series characteristics and rainfall event characteristics, and also rainfall extremes, particularly for long-duration rainfall extremes, which are all important for subsequent streamflow analysis. In addition, the embedded uncertainty of SDRM-MCREM shown in Fig. 5 and Fig. 6 can be used to reflect the internal variability of rainfall.

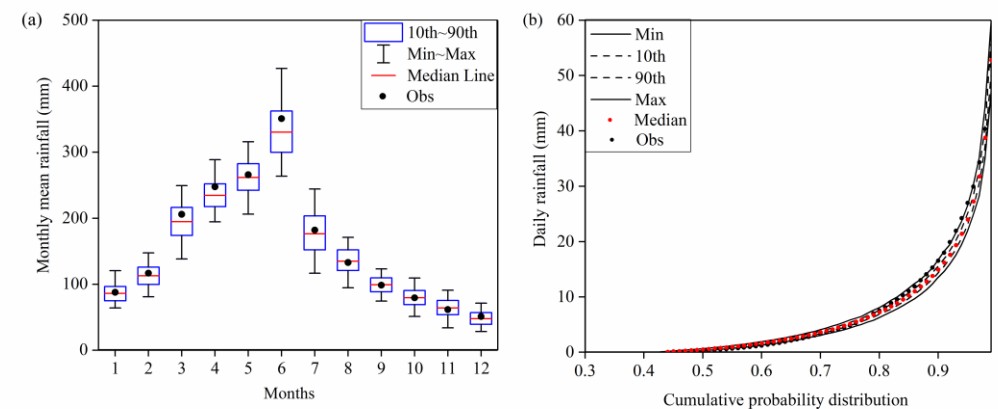


**Figure 5:** Comparison of SDRM-MCREM simulations with observations in (a) monthly mean rainfall and (b) cumulative probability distribution of daily rainfall.

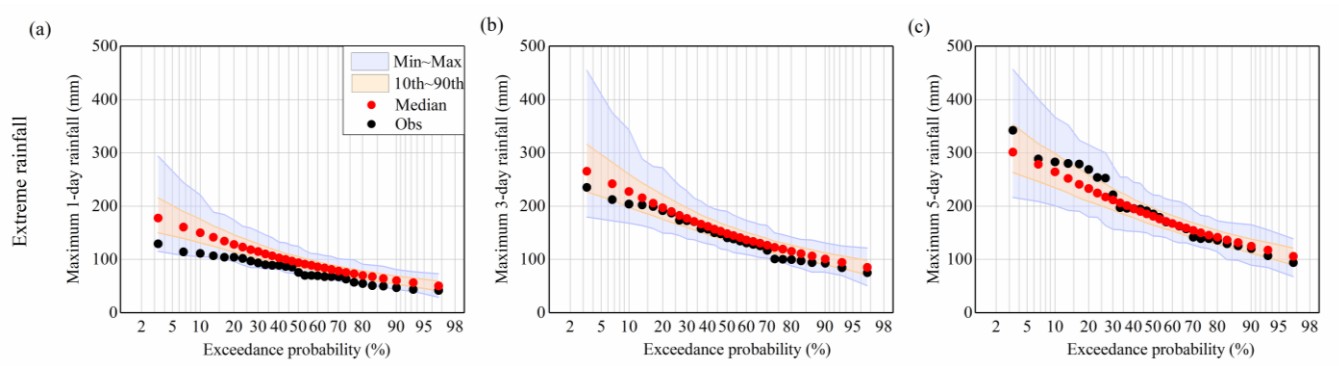





**Figure 6:** Comparison of DRM-MCREM simulations with observations in extreme rainfall, i.e. annual maximum (a) 1-day,   (b) 3-day and (c) 5-day rainfall.

### 4.2.2 Calibration and validation of GR4J model

Based on the method described in Section 3.3, the optimal parameter set of GR4J was selected. The split-sample calibration and validation results with the chosen parameter set are displayed in Table 2. The results in the two rounds show that the GR4J model with the optimum parameter set performs very well, because the values of NS and Y in the calibration and validation periods are mostly above 0.90 and the absolute value of RVE is smaller than 3.2%. The RVE, NS and Y values for the historical period 1971-2000 are -1.72%, 0.93 and 0.92, respectively. Although the GR4J model slightly underestimates peak flows in most cases, this underestimation can to some extent be eliminated as we use the relative change in results in the subsequent analyses.

**Table 2.** Calibration and validation results of GR4J model

| Periods | Y | NS | RVE |
|---|---|---|---|
| The first round | | | |
| Calibration (1971-1990) | 0.90 | 0.91 | -0.54% |
| Validation (1991-2000) | 0.92 | 0.93 | 1.02% |
| The second round | | | |
| Calibration (1981-2000) | 0.93 | 0.94 | -1.06% |
| Validation (1971-1980) | 0.89 | 0.91 | -3.15% |

### 4.3 Overall change in mean and extreme flows under climate change

With the 1,000 realizations of rainfall stochastically simulated using SDRM-MCREM for each bias-corrected GCM rainfall series and one set of PET calculated using bias-corrected temperature data for each GCM to drive the GR4J hydrological model with the selected optimal parameter set, the corresponding 1,000 sets of simulated streamflow for each GCM in the historical period 1971-2000 and the two future periods 2050s and 2080s under four RCP scenarios can be obtained. The influences of climate change on mean flows and extreme flows, including high and low flows, are investigated by means of relative changes (Fig. 7-Fig. 8) in this study. The mean flows are represented by the indices of multi-year monthly and annual average flow. Annual maximum 1-, 3- and 5-day mean flow are adopted to reflect high flows, while low flows are represented by annual minimum 7-, 30- and 90-day mean flow (Gao et al., 2020a). The above annual maximum (or minimum) multi-day mean flow is represented by the highest (or lowest) multi-day average value occurring for any given year (Richter et al., 1996;Kiesel et al., 2019). These adopted indices are all known as hydrological indicators, which have been widely used in hydrological analysis (Olden and Poff, 2003).





To investigate the overall change of mean, high and low flows under climate change, the medians of these indices in the mid-future 2050s and far-future 2080s are compared to that in the historical period 1971-2000. As for high and low flows, the exceedance probabilities of their indices are analysed. The medians in the historical period are extracted from 9,000 sets of simulated streamflow (9 GCMs * 1,000 stochastic simulations), and the medians in the 2050s and 2080s are separately extracted from 36,000 sets of simulated streamflow (4 RCPs * 9 GCMs * 1,000 stochastic simulations). In general, the annual mean flow (Fig. 7) and high flows (Fig. 8) are projected to increase, while the low flows are projected to decline in the future

(Fig. 8). Moreover, the changes in the far-future 2080s are greater than in the mid-future 2050s. For the monthly mean flow, there is an increase in peak-flow seasons (April to June), especially in June, which is probably the main reason for the increase of annual flow. However, the changes in other months are not so consistent, with decreases in July and August but increases in September and December and no significant changes are found in the other months. The annual mean flow is slightly increased by 0.81% in the 2050s and by 3.12% in the 2080s, respectively. For high flows, the changes of annual maximum 1-,

3- and 5-day mean flow in the 2080s increases with the increase of return periods while the changes in the 2050s become stable when the exceedance probability is smaller than 70%. Furthermore, with the longer duration of high flows, the magnitude of increase decreases. However, for the low flows an opposite result is found. The changes of annual minimum 7-, 30- and 90-day mean flow decline with increasing return periods, but rise with the longer duration of low flows, except for the annual minimum 90-day mean flow with a return period smaller than 2 years (i.e. exceedance probability larger than 50%). This

phenomenon can be explained by that there is a larger probability for longer duration of low flows to contain large flow, particularly for the annual minimum 90-day mean flow with small return periods. The larger flow contained in the annual minimum 90-day mean flow may present an increase in the future that would probably lead to the decline in the magnitude of decrease. The detailed changes of high and low flows at specific return periods, i.e. 5, 10 and 20 years in the future are shown in Table 3. High flows with a return period of 20 years increase with around 11% in the 2050s and approximately 25% in the

2080s, while low flows with the same return period decrease with 9-11% and 10-12% in the 2050s and 2080s, respectively.

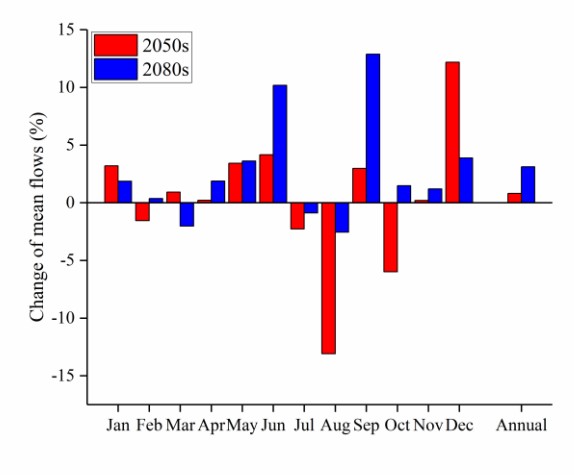





**Figure 7:** Change of mean flows in the 2050s and 2080s relative to the historical period 1971-2000.

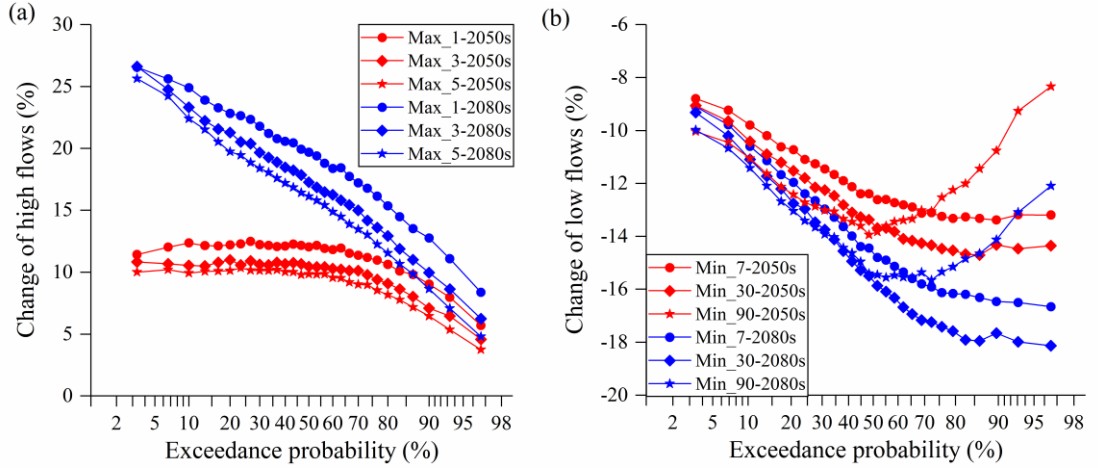

**Figure 8.** Change of (a) high flows and (b) low flows in the 2050s and 2080s relative to the historical period 1971-2000. The symbols Max_1, Max_3 and Max_5 represent the annual maximum 1-, 3- and 5-day mean flow, while Min_7, Min_30 and Min_90 represent the annual minimum 7-, 30- and 90-day mean flow, respectively.

**Table 3.** Future change of high flows and low flows at specific return periods

| Hydrological indicators | 5-year return period | | 10-year return period | | 20-year return period | |
|---|---|---|---|---|---|---|
| | 2050s | 2080s | 2050s | 2080s | 2050s | 2080s |
| High flows | | | | | | |
| Annual maximum 1-day flow | 11.7% | 22.3% | 12.2% | 24.4% | 11.6% | 25.1% |
| Annual maximum 3-day mean flow | 10.8% | 21.1% | 11.0% | 23.3% | 10.9% | 25.1% |
| Annual maximum 5-day mean flow | 10.4% | 20.3% | 10.7% | 22.9% | 10.4% | 24.5% |
| Low flows | | | | | | |
| Annual minimum 7-day mean flow | -11.5% | -13.1% | -10.1% | -11.9% | -9.0% | -10.3% |
| Annual minimum 30-day mean flow | -12.3% | -14.1% | -10.8% | -12.7% | -9.5% | -11.0% |
| Annual minimum 90-day mean flow | -13.0% | -15.0% | -11.7% | -13.6% | -10.6% | -12.1% |

**4.4 Contribution of different uncertainty sources for mean and extreme flows**

Since the uncertainty of streamflow in the historical period can be largely attributed to internal climate variability, only the contribution of different uncertainty sources, i.e. from RCPs, GCMs and internal climate variability, in the future periods is shown (Fig. 9-Fig. 13) in this study. For mean flows, the uncertainty stemming from GCMs dominates for all months except October in the 2050s, followed by internal climate variability and RCP uncertainty (Fig. 9). Specifically, GCMs contributes





more than 70% to the total uncertainty in annual mean flow. Compared to the 2050s, the effect of RCP and GCM uncertainty

in the 2080s in general becomes larger throughout the year, although the contribution of GCM uncertainty in the annual mean

flow shows a slight decrease. The fraction of internal climate variability consistently decreases in the 2080s compared to the

2050s.

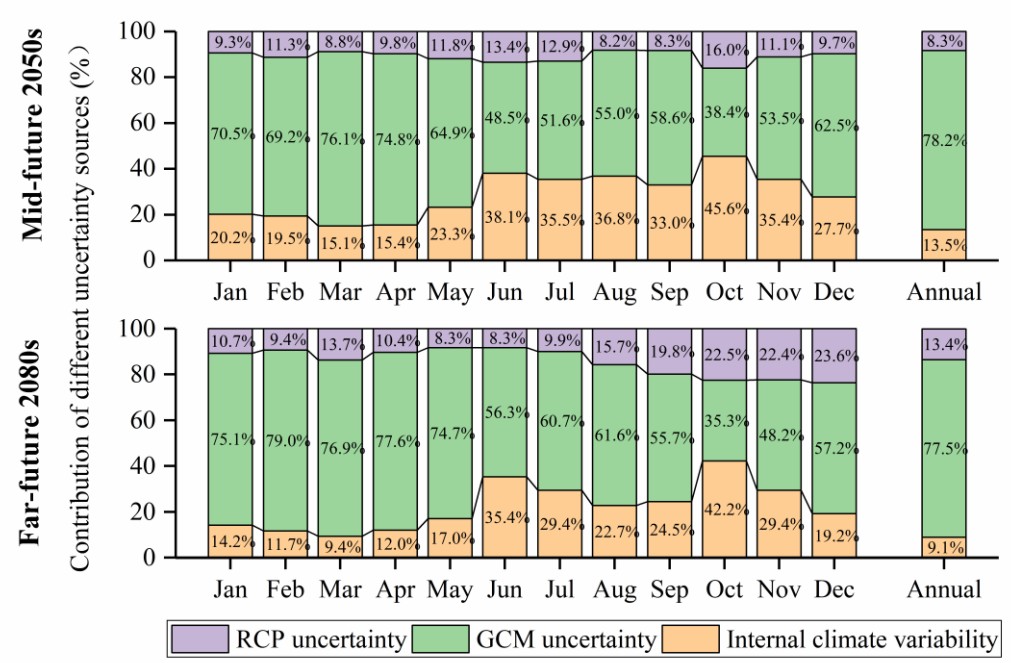

**Figure 9:** Contribution of three uncertainty sources, i.e. from RCPs, GCMs and internal climate variability for mean flows in the 2050s and

2080s, respectively.

For high flows, internal climate variability and GCM uncertainty are clearly the major sources for the two future periods,

with internal climate variability becoming dominant when return periods are larger than four years or smaller than 1.1 years

(Fig. 10). The contribution of RCP uncertainty in the 2050s increases with increasing return periods, while it is constant for

all return periods in the 2080s. Similar results are found for annual maximum 1-, 3- and 5-day mean flow. Fig. 11 presents the

contribution of the three uncertainty sources for high flows at return periods of 5, 10 and 20 years. Firstly, compared to the

2050s, internal climate variability and GCM uncertainty become even more pronounced and RCP uncertainty is less relevant

in the 2080s. Secondly, with increasing return periods, internal climate variability contributes a larger fraction while GCM

uncertainty contributes a smaller fraction. Lastly, the effect of RCP and GCM uncertainty becomes gradually obvious with the

longer duration of high flows while internal climate variability shows the opposite behavior.





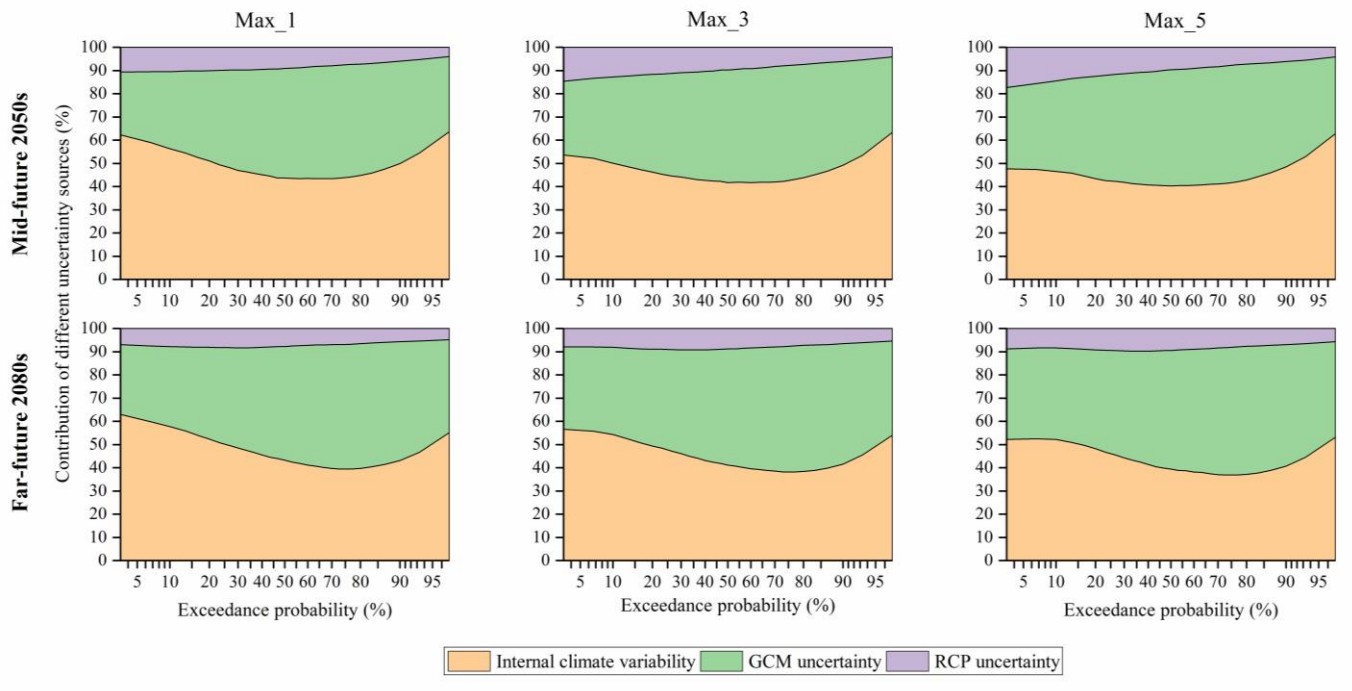

**Figure 10:** Contribution of the three uncertainty sources for high flows in the 2050s and 2080s, respectively. Symbols Max_1, Max_3 and Max_5 represent the annual maximum 1-, 3- and 5-day mean flow, respectively.



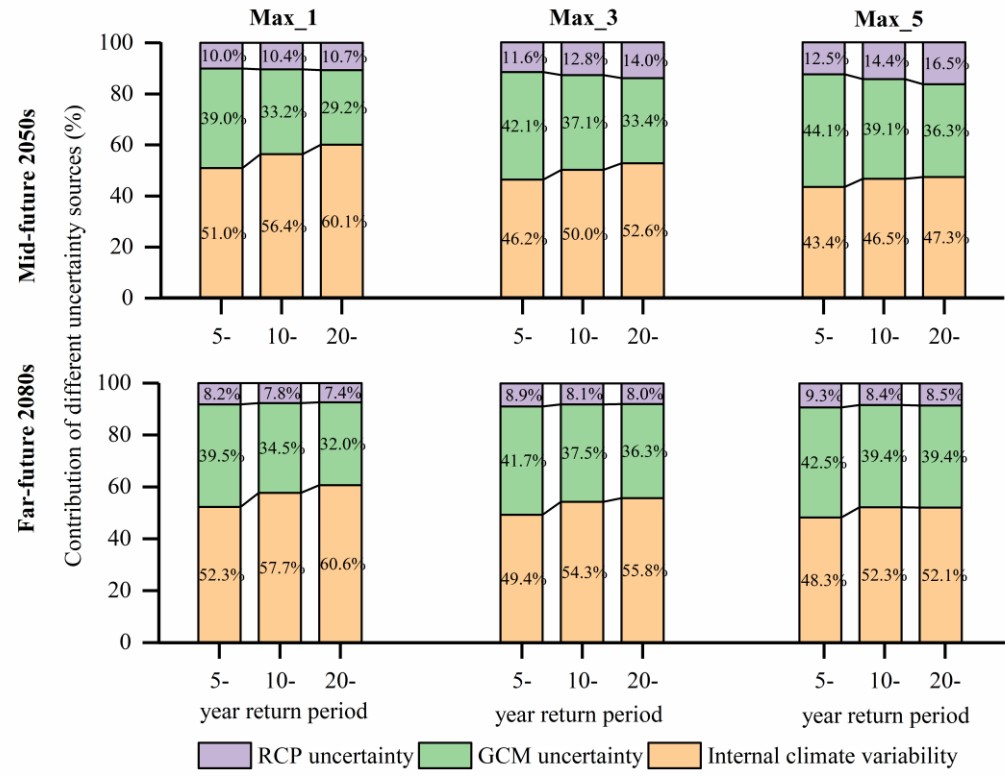

**Figure 11.** Contribution of the three uncertainty sources for high flows at 5-, 10- and 20-year return periods in the 2050s and 2080s, respectively. Symbols Max_1, Max_3 and Max_5 represent the annual maximum 1-, 3- and 5-day mean flow, respectively. The figures on the bars represent the contribution values of different uncertainty sources.

However, different from the findings for high flows, the role of RCP uncertainty is more evident for low flows (Fig. 12). Especially in the far future, the uncertainty contributed by RCPs increases greatly, while the uncertainty due to internal climate variability reduces significantly. With longer projection horizons, GCM uncertainty also contributes a larger fraction to the total uncertainty and becomes the dominant factor in the 2080s. In addition, with the longer duration of low flows, the effect of internal climate variability becomes larger. This is possibly because there is a larger probability for longer duration of low flows to contain large flow and these large flows are greatly influenced by internal climate variability. Fig. 13 gives the detailed contribution of the three uncertainty sources for low flows at 5-, 10- and 20-year return periods.





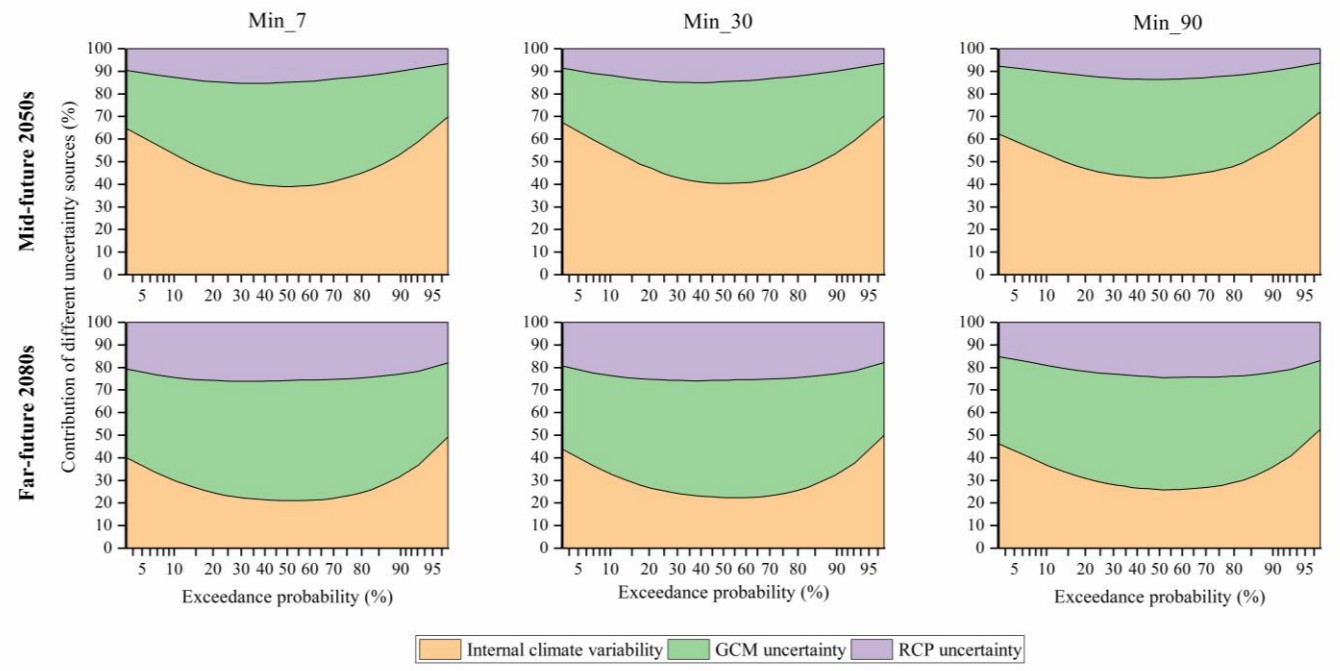

**Figure 12:** Contribution of the three uncertainty sources for low flows in the 2050s and 2080s, respectively. Symbols Min_7, Min_30 and Min_90 represent the annual minimum 7-, 30- and 90-day mean flow, respectively.



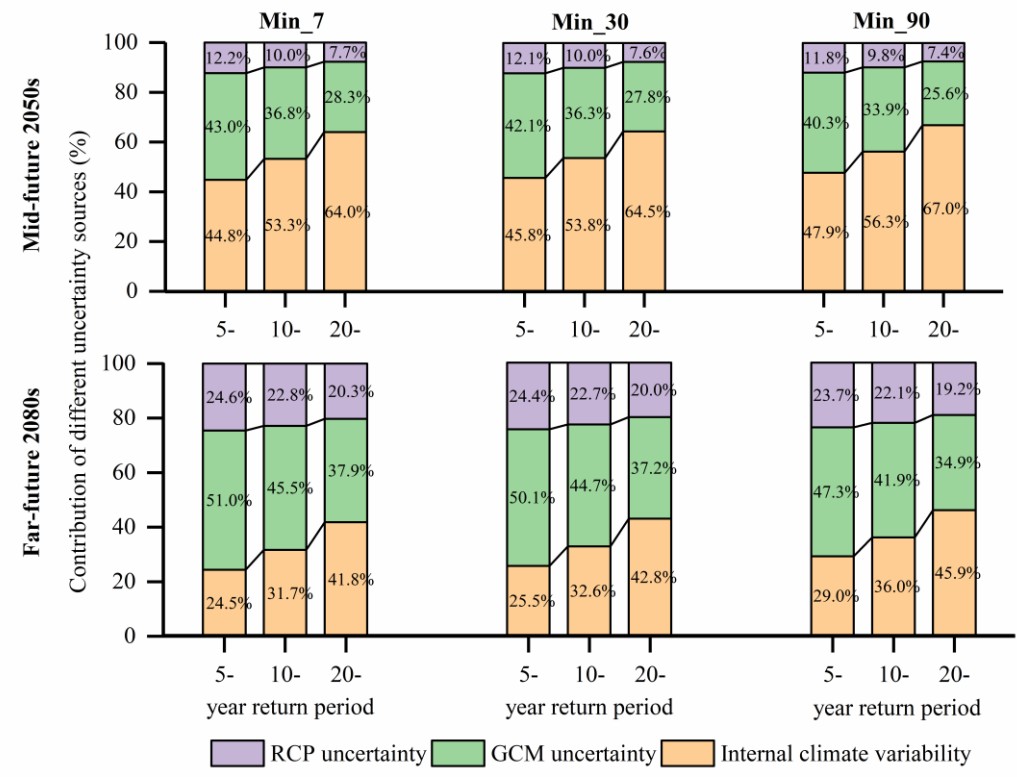

**Figure 13:** Contribution of the three uncertainty sources for low flows at 5-, 10- and 20-year return periods in the 2050s and 2080s, respectively. Symbols Min_7, Min_30 and Min_90 represent the annual minimum 7-, 30- and 90-day mean flow, respectively. The figures on the bars represent the contribution values of different uncertainty sources.

Generally, the contribution of different uncertainty sources varies with streamflow characteristics, i.e. high, mean and low flows. The link between the relative importance of different uncertainty sources for flows and those for climate variables (i.e. rainfall and temperature) will be discussed in detail in Section 5.

## 5 Discussion

Projected streamflow changes shown in Fig. 7-8 indicate that the Qu River basin will probably have more water resources on an annual basis, but hydrological extreme events like high flows and low flows will become more serious in the mid-future period and particularly the far-future period. Both wet-seasonal and annual flows are expected to experience an increase in the future, keeping consistent with the increase of rainfall, but the change in streamflow is smaller than that in rainfall (by comparing Fig. 4 and Fig. 7). This is mainly due to the rise of temperatures (Fig .4) that leads to more water loss in the form of evapotranspiration. In addition, there is a clear tendency toward a higher flood and drought risk in this study area (Fig. 8). This finding is in agreement with previous studies in the same river basin (Gao et al., 2020c), and also further confirms the





findings regarding changes of extreme rainfall event characteristics by Gao et al. (2020b) that are highly associated with the occurrence of floods and droughts. Gao et al. (2020b) concluded that the frequency of extreme dry spell events and extreme rainfall events will probably be projected to increase, implying that the Qu River basin will have a larger possibility to experience droughts and floods in the future. Furthermore, the rainfall temporal patterns are becoming more-centralized, particularly for the peak-delayed rainfall types and extreme rainfall events, which will result in larger peak flows and probably induce floods in this study area, especially in the late 21st century (Gao et al., 2020b). The above findings can provide important implications for water management in the Qu River basin. For example, this can guide water managers to store more excessive water in wet seasons and release water in dry seasons by appropriate regulation of reservoirs to alleviate flood and drought risks.

Uncertainty in projected streamflow is the result of uncertainty in projected climate variables, i.e. rainfall and temperature, and the relative importance of different uncertainty sources may vary with the variable of interest and given projection time. Fig. S1-S4 present the contribution of different uncertainty sources for rainfall and temperature. In this study, we did not take the internal variability of temperature into account, but this would not make a big difference. This is because most studies pointed out that the fraction of variance attributed to internal variability of the temperature is negligible when compared to scenario uncertainty and model uncertainty, especially in the second half of 21st century (Lafaysse et al., 2014;Hingray and Saïd, 2014;Fatichi et al., 2016). For rainfall (Fig. S1-S2), it can be clearly seen that the internal variability is the largest uncertainty source that can account for approximately 54-60% on average, followed by GCM uncertainty taking up about 32-35% and RCP uncertainty 8-11%, both in the 2050s and 2080s. However, in terms of temperatures (Fig. S3-S4), GCM uncertainty evidently prevails in the 2050s, contributing to more than 70% of the total uncertainty in daily maximum and mean temperatures, while RCP uncertainty begins to play a more significant role than GCM uncertainty in the 2080s. The effect of RCP uncertainty for daily minimum temperature is in general larger than that for daily maximum and mean temperatures. The above findings that both GCM uncertainty and internal climate variability contribute substantially to the overall uncertainty in rainfall changes and RCP uncertainty is the primary source for air temperature, particularly for far future projections, are consistent with previous climate impact studies (Hawkins and Sutton, 2009;Schewe et al., 2014;Yip et al., 2011). Due to the combined effects of rainfall and temperature on the streamflow formation, the impacts of RCP and GCM uncertainty on streamflow are larger than those for rainfall. This phenomenon becomes more pronounced for the mean flow and low flows. For example, the uncertainty due to internal climate variability and GCM uncertainty for high flows have become comparable, i.e. both accounting for about 44-48%, and RCP uncertainty accounts for about 10% of the total uncertainty in the future. Particularly for low flows, the contribution of GCM uncertainty and RCP uncertainty in the 2050s and 2080s has increased from 40% and 13% to 48% and 24%, respectively, and internal climate variability has decreased from 47% to 28% on average. Generally, it can be summarized that the uncertainty in high flows is still mainly influenced by that of rainfall while the uncertainty in low flows is primarily affected by temperatures, especially in the late 21st century. The relative importance of the three uncertainty sources for high flows found in this study is in agreement with the conclusions by Kay et al. (2008) and





Jung et al. (2011). Nevertheless, since the number of similar studies focusing on different kinds of flows is limited at present, the different results found for mean, high and low flows in this study need further comparison and verification.

Although this study mainly concentrated on investigating how the uncertainty of climate projections is propagated in streamflow projections, there are other sources of uncertainty, like uncertainties originating from downscaling methods, hydrological model structures and hydrological parameters, etc. Chen et al. (2011) and Meaurio et al. (2017) found that downscaling methods might also have a large contribution to the uncertainty in peak-flow projections, as different types of downscaling methods might lead to significantly different extreme high flows, and uncertainty in simulated extreme low flows

is also critically impacted by hydrological model structures as well as calibration strategies (De Niel et al., 2019;Vansteenkiste et al., 2014;Velázquez et al., 2013). Therefore, to obtain a comprehensive insight into projected changes of high flows and low flows and the uncertainty therein, all sources of uncertainty arising from scenarios, climate models, internal climate variability, downscaling methods, hydrological models and hydrological parameters can be considered in future studies.

## 6 Conclusions

In this study, we investigated the streamflow changes of the Qu River basin in the mid-future period 2050s and the far-future period 2080s relative to the historical period 1971-2000. The contribution of three different sources of uncertainty, i.e. RCP uncertainty, GCM uncertainty and internal climate variability was also quantified by using four scenarios, nine GCMs and 1,000 simulations of SDRM-MCREM (a stochastic daily rainfall model coupling a Markov chain model with a rainfall event model). The conclusions can be summarized as follows:

(1) Annual mean flow and wet seasonal flow (April-June) are projected to increase in the two future periods, and hydrological extreme events are becoming more extreme, which are indicated by that high flows become higher and low flows become lower, especially for the 2080s. This indicates that the Qu River basin will be probably faced with a higher risk of floods and droughts in the future.

(2) For mean flows, GCM uncertainty is generally the largest contributor to the total uncertainty, followed by internal climate
variability and RCP uncertainty. The effect of GCM and RCP uncertainty on mean flows in the 2080s is generally greater than that in the 2050s.

(3) For high flows, internal climate variability and GCM uncertainty play a comparable role in the total uncertainty and are the two major uncertainty sources. The uncertainty of high flows is mainly affected by that of rainfall extremes. The relative importance of different uncertainty sources for rainfall extremes from large to small are internal climate variability,
GCM uncertainty and RCP uncertainty. For temperature, GCM uncertainty is dominant in the 2050s but RCP uncertainty gradually becomes dominant in the 2080s.

(4) The role of RCP and GCM uncertainty is more pronounced for future low flow projections. Especially in the 2080s, GCM uncertainty has become the dominant uncertainty source, and RCPs have almost the same contribution to the total uncertainty as internal climate variability. The uncertainty of low flows is primarily influenced by the uncertainty in

470        temperature so that the effect of internal climate variability is much smaller than for high flows.

**Supplements**

The supplement can be downloaded from the following LINK.

**Code and data availability**

A detailed simulation procedure of the rainfall event model using in the stochastic daily rainfall model SDRM-MCREM can
be found from https://doi.org/10.1016/j.jhydrol.2018.06.073. The observed streamflow data and rainfall data from Zhejiang
Bureau of Hydrology are available upon request from the corresponding author (yuepingxu@zju.edu.cn). The observed
temperature data can be downloaded from https://data.cma.cn/. The GCM simulated rainfall and temperature data can be
obtained from https://esgf-node.llnl.gov/search/cmip5/.

**Author contributions**

CG, MB, and Y-P Xu designed this study together. CG conducted the modelling, the analysis and preparation of the manuscript
with support from all co-authors. All co-authors largely contributed to the discussion of the results and revising the paper.

**Competing interests**

The authors declare that they have no conflict of interest.

**Acknowledgements**

This study is financially supported by the National Key Research and Development Plan "Inter-governmental Cooperation in
International Scientific and Technological Innovation" (2016YFE0122100) and the National Natural Science Foundation of
China (91547106). We sincerely acknowledge Zhejiang Bureau of Hydrology and the National Climate Centre of China
Meteorological Administration for providing us meteorological and hydrological data used in this study.

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
