# Peer review of "Assessment of extreme flows and uncertainty under climate change: disentangling the contribution of RCPs, GCMs and internal climate variability"

_Hydrology and Earth System Sciences, 2020_

## Referee Comment (RC1) · Anonymous Referee #1 · 4 Mar 2020

This manuscript aims to investigate the future changes of mean flows, high flows and low flows under climate change and, in particular, to quantify the contribution of three uncertainty sources from RCPs, GCMs and internal climate variability in these different flows. The innovative thing is that the internal climate variability is mainly reflected by using simulations of the stochastic rainfall model SDRM-MCREM developed by the authors. ANOVA is adapted to estimate the contributions all these uncertainties. The contents of this paper are therefore interesting and fall within the scope of HESS. Although there are no critical problems, there are a few issues that need to be considered and discussed before recommending this paper for publication. 1. Since the internal climate variability in this paper is represented by the simulations of SDRM-MCREM, whether

the contribution of internal climate variability to the total uncertainty is directly relevant with the performances of SDRM-MCREM? For example, in Figure 10, the contribution of internal climate variability in annual maximum 1-day flow for larger return periods is obviously larger than smaller return periods, whether this indicates the poor performance of SDRM-MCREM in simulating extremes? Please explain. 2. Another main usage of stochastic rainfall model is to downscale climate model outputs by adjusting parameters of stochastic rainfall models for climate change impact studies. The future GCMs rainfall data in this study are directly simulated by SDRM-MCREM using the bias corrected GCM future data rather than through downscaling by SDRM-MCREM. Can the authors explain why you conducted like this? 3. Obviously, there also exists uncertainty in the process of hydrological modelling. Why did your study only consider the uncertainty of RCPs, GCMs and internal climate variability and neglects the uncertainty of hydrological parameters that seems can be easily incorporated. Please explain.

Some other minor points to consider are listed below: L36. "responses of" <-> "responses to". L39. "the coupled system" – the atmosphere-ocean coupled system? Please make it clear. L48. "The relative importance" refers to what? Please make it clear. L124. "In this study, we used the distribution mapping (DM) method to correct GCM-simulated climate variable" – at this point in the text, some further explanation about why choosing the DM method is needed in the context. L129. Please check the correctness of Eq. (2). L189. "is" <-> "was". The tense of this paper in the method part is a bit confusing. Please check the whole paper and ensure proper use of the tense. L338. In this paper, when investigating the changes of high flows and low flows, the 5-, 10- and 20-year return periods are adopted. Why not use the larger return periods such 50-year and 100-year return periods that are more useful information for assessment of extreme hydrological events? L421. "account for approximately 54-60% on average" – Does this mean the proportion of the total uncertainty? Please make it clear.

In summary, the manuscript is well-structured, and the methods used and the results

are interesting and useful. I hope this manuscript can be accepted in publication in HESS after minor or moderate revision.

СЗ

---

## Referee Comment (RC2) · Anonymous Referee #2 · 12 Mar 2020

Projections of future climate change impacts on streamflow, particularly of extreme flows are beneficial for decision makers in future water resources management and development of adaptation strategies of flood and drought control. However, these projections are associated with large uncertainties from various sources. This manuscript predicted future streamflow changes in the Qu River basin in China and quantified uncertainty sources such as Representative Concentration Pathways (RCPs), Global Climate Models (GCMs), and internal climate variability, using the analysis of variance (ANOVA) to quantify the contribution of different sources. The experiments were designed appropriately and the results were well interpreted. The manuscript was well written. It can be accepted with minor improvements. Here below are a few suggestions:

1) This manuscript only explained how to disaggregate monthly precipitation into daily precipitation considering the internal climate variability. How to downscale air temperature from monthly scale to daily scale should be also explained in the manuscript. 2) Section 4.2.2: Since this paper focuses on extreme flow projections, please add a short paragraph on the performance of the hydrological model in extreme high and low flow simulations in the historical period.

Minor: 1) Figure 2: Replace 'dry spell' with 'Dry spell'. 'Modium' in the table should be 'Medium'. 2) Figure 3 is not clear enough. Please enlarge the figures to improve the readability. 3) Figure 4: In each sub-figure, there is a small plot box with three curves. Please denote the plot box in the figure caption. 4) Figure 6: Replace 'DRM-MCREM' with 'SDRM-MCREM'. 5) Line 139: 'he features' should be 'the features'.

---

## Author Comment (AC2) · 29 Apr 2020

Dear Referee #2,

We highly appreciate your review and useful suggestions for our manuscript. We provide our answers to your queries below.

Kind regards, all authors

Queries by anonymous referee #2 RC2 & answers by authors are as follows:

Comment #1: This manuscript only explained how to disaggregate monthly precipitation into daily precipitation considering the internal climate variability. How to downscale air temperature from monthly scale to daily scale should be also explained in the manuscript.

Authors' response: Thanks for your comments. In this study, we did not disaggregate monthly rainfall into daily rainfall, because we directly used the daily rainfall and temperature data of GCMs downloaded from https://data.cma.cn/ as described in Section 2. In fact, we spatially downscaled GCM simulated rainfall and temperature data in the historical and future periods rather than temporally downscaled the data, i.e. downscaling from the coarse GCM spatial resolution to the catchment resolution. This has been done using the distribution mapping (DM) method. The detailed downscaling methods are described in Section 3.1. In addition, this study indeed only considered the internal variability of rainfall through directly generating realizations of rainfall using the developed stochastic rainfall model SDRM-MCREM, the internal variability of temperature is not taken into consideration. Considering that the internal variability of rainfall is always large and comparable to or even greater than the recognized large uncertainty of GCMs (Hingray and Saïd, 2014; Giorgi, 2002), it should not be neglected in the analysis of different uncertainty sources in rainfall projections. Regarding the internal variability of temperature, several studies pointed out that it is usually very small compared with other uncertainty sources of temperature like GCM uncertainty and RCP uncertainty and therefore can be ignored without large consequences (Lafaysse et al., 2014; Hingray and Saïd, 2014; Fatichi et al., 2016). For that reason we did not consider the internal variability of temperature in this study as an uncertainty source and this has also been explained in the discussion section (see Page 22, Line 416-420).

References:

Hingray, B., and Saïd, M.: Partitioning Internal Variability and Model Uncertainty Components in a Multimember Multimodel Ensemble of Climate Projections, J. Clim., 27, 6779-6798, https://doi.org/10.1175/jcli-d-13-00629.1, 2014.

Giorgi, F.: Dependence of the surface climate interannual variability on spatial scale,
Geophys. Res. Lett., 29, 16-11-16-14, https://doi.org/10.1029/2002gl016175, 2002.

Lafaysse, M., Hingray, B., Mezghani, A., Gailhard, J., and Terray, L.: Internal variability and model uncertainty components in future hydrometeorological projections: The Alpine Durance basin, Water Resour. Res., 50, 3317-3341, 2014.

Fatichi, S., Ivanov, V. Y., Paschalis, A., Peleg, N., Molnar, P., Rimkus, S., Kim, J., Burlando, P., and Caporali, E.: Uncertainty partition challenges the predictability of vital details of climate change, Earth Future, 4, 240-251, https://doi.org/10.1002/2015ef000336, 2016.

Comment #2: Section 4.2.2: Since this paper focuses on extreme flow projections, please add a short paragraph on the performance of the hydrological model in extreme high and low flow simulations in the historical period.

Authors' response: Thank you for your useful suggestion. We will add a relevant figure and some information about the performance of the GR4J hydrological model with the selected optimum parameter set in reproducing extreme flows including high flows and low flows in Section 4.2.2.

Comment #3: Figure 2: Replace 'dry spell' with 'Dry spell'. 'Modium' in the table should be'Medium'.

Authors' response: Thank you. These will be modified accordingly, i.e. "dry spell" being replaced with "Dry spell" and "Modium" replaced with "Medium" in Figure 2.

Comment #4: Figure 3 is not clear enough. Please enlarge the figures to improve the readability.

Authors' response: Thank you. We will redraw Figure 3 to make it much clearer to read.

Comment #5: Figure 4: In each sub-figure, there is a small plot box with three curves. Please denote the plot box in the figure caption.

[Figure]

Authors' response: Thank you very much. We will add a description for these small plot boxes to the caption of Figure 4. The adjusted caption of Figure 4 will read "Ensemble averages of GCM simulated monthly mean (a) rainfall, (b) daily maximum temperature, (c) daily minimum temperature and (d) daily mean temperature for the historical and two future periods under four RCPs after bias correction. The small graph in each sub-figure shows the ensemble averages of all GCMs and RCPs for the corresponding variable in the historical period and the two future periods, i.e. 2050s and 2080s."

Comment #6: Figure 6: Replace 'DRM-MCREM' with 'SDRM-MCREM'.

Authors' response: Sorry for the mistake. We will replace "DRM-MCREM" with "SDRM-MCREM' in the caption of Figure 6.

Comment #7: Line 139: 'he features' should be 'the features'.

Authors' response: Thank you. We will modify "he features" to "the features".

---

## Author Response (AR1)

Dear Editor and reviewers,

We highly appreciate your review and useful suggestions for our manuscript. We have made corresponding revisions in our manuscript according to your comments. Hopefully these changes will fulfill your expectations. Below are the point-to-point responses to the comments. The comments from reviewers are in blue while our responses appear in black. The red color indicates the newly-added contents in the revised manuscript. The page and line numbers given below refer to the numbers in the version of revised manuscript without changes tracked.

Yours truthfully,

Kind regards, all authors

**Referee #1:**

**Comment #1:** Since the internal climate variability in this paper is represented by the simulations of SDRM-MCREM, whether the contribution of internal climate variability to the total uncertainty is directly relevant with the performances of SDRM-MCREM? For example, in Figure 10, the contribution of internal climate variability in annual maximum 1-day flow for larger return periods is obviously larger than smaller return periods, whether this indicates the poor performance of SDRM-MCREM in simulating extremes? Please explain.

**Authors' response:** Thanks for your question. There are two commonly-used approaches for the quantification of the internal variability of the climate system: the first one is using multiple members of GCMs to reflect internal climate variability (Bracegirdle et al., 2014); the second one is taking the randomness of weather generators or stochastic rainfall models as the internal climate variability (Lafaysse et al., 2014; Fatichi et al., 2016). In this way, it is unavoidable that the contribution of internal climate variability is dependent on the performances of the adopted methods. This means that the contribution of internal climate variability to the total uncertainty is not only directly dependent on the performance of SDRM-MCREM in this study, but also dependent on the performance of other methods if other methods were adopted. In this study, to avoid the impacts of a poor performance of a stochastic rainfall model on its randomness and affecting the representation of internal climate variability, we have made great efforts to apply the well-performing stochastic rainfall model SDRM-MCREM, which showed good results for simulation of both rainfall time-series characteristics and rainfall event characteristics. The advantages of SDRM-MCREM compared to other weather generators and stochastic rainfall models are described in Gao et al. (2020a). In addition, the inferior performance of weather generators and stochastic rainfall models in simulating extremes and inherent large uncertainties is a common problem. Compared to other weather

generators, the SDRM-MCREM performs relatively better in reproducing rainfall extremes (Gao et al. 2020a). To draw conclusions about the contributions of different uncertainty sources in a more accurate and reliable way in this study, we summarized the findings of this study for an average return period and also compared these results with previous studies (see Page 24, Line 450-457).

References:

Bracegirdle, T. J., Turner, J., Hosking, J. S., and Phillips, T.: Sources of uncertainty in projections of twenty-first century westerly wind changes over the Amundsen Sea, West Antarctica, in CMIP5 climate models, Clim. Dyn., 43, 2093-2104, https://doi.org/10.1007/s00382-013-2032-1, 2014.

Lafaysse, M., Hingray, B., Mezghani, A., Gailhard, J., and Terray, L.: Internal variability and model uncertainty components in future hydrometeorological projections: The Alpine Durance basin, Water Resour. Res., 50, 3317-3341, 2014.

Fatichi, S., Ivanov, V. Y., Paschalis, A., Peleg, N., Molnar, P., Rimkus, S., Kim, J., Burlando, P., and Caporali, E.: Uncertainty partition challenges the predictability of vital details of climate change, Earth Future, 4, 240-251, https://doi.org/10.1002/2015ef000336, 2016.

Gao, C., Booij, M. J., and Xu, Y.-P.: Development and hydrometeorological evaluation of a new stochastic daily rainfall model: coupling Markov chain with rainfall event model, J. Hydrol., under revision, 2020a.

**Comment #2:** Another main usage of stochastic rainfall model is to downscale climate model outputs by adjusting parameters of stochastic rainfall models for climate change impact studies. The future GCMs rainfall data in this study are directly simulated by SDRM-MCREM using the bias corrected GCM future data rather than through downscaling by SDRM-MCREM. Can the authors explain why you conducted like this?

**Authors' response:** There are two reasons why we conducted the study in this way. Firstly, the bias-corrected future GCM rainfall data already contain sufficient information to reflect the impacts of climate change on rainfall characteristics. Gao et al. (2020b) investigated the changes of rainfall event characteristics using bias-corrected historical and future GCM data, and found that not only the distributions of rainfall duration and rainfall depth would change, but also the temporal patterns of rainfall events would change in the future. Secondly, obtaining realizations of future rainfall time series through simulating the bias-corrected future GCM data using SDRM-MCREM in order to consider future internal climate variability is more straightforward and easier. As far as we know, using weather generators or stochastic rainfall models to downscale GCM future simulations currently is mainly through perturbing the parameters of weather generators, like the transition probabilities of rainfall occurrence and parameters of the distribution of rainfall amount, using monthly averaged additive or multiplicative change factors of GCM projections (Chen et al., 2012; Li and Babovic, 2018). However, it cannot be guaranteed that these kinds of downscaling methods can fully incorporate changes of rainfall characteristics (e.g.

temporal patterns of rainfall events) although it has been commonly used in previous studies. In addition, it is complicated to carry out the whole downscaling process. Based on the above, it is considered more convenient and useful to directly simulate the bias-corrected GCM rainfall series using SDRM-MCREM in this study.

References:

Gao, C., Booij, M. J., and Xu, Y. P.: Impacts of climate change on characteristics of daily-scale rainfall events based on nine selected GCMs under four CMIP5 RCP scenarios in Qu River basin, east China, Int. J. Climatol., 40, 887-907, https://doi.org/10.1002/joc.6246, 2020b.

Chen, J., Brissette, F. P., and Leconte, R.: Downscaling of weather generator parameters to quantify hydrological impacts of climate change, Climate Research, 51, 185-200, https://doi.org/10.3354/cr01062, 2012.

Li, X., and Babovic, V.: Multi-site multivariate downscaling of global climate model outputs: an integrated framework combining quantile mapping, stochastic weather generator and Empirical Copula approaches, Clim. Dyn., 52, 5775-5799, https://doi.org/10.1007/s00382-018-4480-0, 2018.

**Comment #3:** Obviously, there also exists uncertainty in the process of hydrological modelling. Why did your study only consider the uncertainty of RCPs, GCMs and internal climate variability and neglects the uncertainty of hydrological parameters that seems can be easily incorporated. Please explain.

**Authors' response:** Thanks for your question. In the process of hydrological modelling, uncertainty in hydrological model structures is also present besides uncertainty in hydrological parameters. There are two reasons why we did not take the uncertainty in hydrological modelling into consideration in this study. Firstly, many previous studies indicated that uncertainty originating from climate projections is generally larger than uncertainty in the hydrological simulation process (Teng et al., 2012;Karlsson et al., 2016), and uncertainty from hydrological model structures and parameters sets is less important for peak flows (Vetter et al., 2016;De Niel et al., 2019) (See Page 2, Line 56-63). Repeating their work seems not very necessary. In addition, the main purposes of this study are (1) to use the newly-developed well-performing stochastic rainfall model SDRM-MCREM to generate multiple realizations of GCM data and reflect internal climate variability; (2) to investigate how climate projection uncertainties, including RCP uncertainty, GCM uncertainty and internal climate variability, propagate into streamflow projections and estimate their contributions to streamflow projection uncertainty. This has also been described in the introduction section (see Page 3, Line 88-96). Because of this, we did not take the uncertainty of hydrological modelling into account in this study. For future research work, to obtain a comprehensive insight into projected changes of high flows and low flows and the contributions of different uncertainty sources, it is aimed to consider all sources of uncertainty arising from scenarios, climate models, internal climate variability, downscaling methods, hydrological models and hydrological parameters (See Page 25, Line 466-468).

References:

De Niel, J., Van Uytven, E., and Willems, P.: Uncertainty Analysis of Climate Change Impact on River Flow Extremes Based on a Large Multi-Model Ensemble, Water Resour. Manag., 33, 4319-4333, https://doi.org/10.1007/s11269-019-02370-0, 2019.

Karlsson, I. B., Sonnenborg, T. O., Refsgaard, J. C., Trolle, D., Børgesen, C. D., Olesen, J. E., Jeppesen, E., and Jensen, K. H.: Combined effects of climate models, hydrological model structures and land use scenarios on hydrological impacts of climate change, J. Hydrol., 535, 301-317, https://doi.org/10.1016/j.jhydrol.2016.01.069, 2016.

Teng, J., Vaze, J., Chiew, F. H. S., Wang, B., and Perraud, J.-M.: Estimating the Relative Uncertainties Sourced from GCMs and Hydrological Models in Modeling Climate Change Impact on Runoff, J. Hydrometeorol., 13, 122-139, https://doi.org/10.1175/jhm-d-11-058.1, 2012.

Vetter, T., Reinhardt, J., Flörke, M., van Griensven, A., Hattermann, F., Huang, S., Koch, H., Pechlivanidis, I. G., Plötner, S., Seidou, O., Su, B., Vervoort, R. W., and Krysanova, V.: Evaluation of sources of uncertainty in projected hydrological changes under climate change in 12 large-scale river basins, Clim. Change, 141, 419-433, https://doi.org/10.1007/s10584-016-1794-y, 2016.

**Comment #4:** L36. "responses of" <-> "responses to".

**Authors' response:** Thank you. We have replaced "responses of" with "responses to".

**Comment #5:** L39. "the coupled system" – the atmosphere-ocean coupled system?

Please make it clear.

**Authors' response:** Thank you. "the coupled system" does refer to "the coupled atmosphere-ocean system". We have revised it.

**Comment #6:** L48. "The relative importance" refers to what? Please make it clear.

**Authors' response:** Thank you. Here "The relative importance" refers to the relative importance of different uncertainty sources. Therefore, the original content has been modified to "The relative importance of different uncertainty sources", which can be found in Page 2, Line 48

**Comment #7:** L124. "In this study, we used the distribution mapping (DM) method to correct GCM-simulated climate variable" – at this point in the text, some further explanation about why choosing the DM method is needed in the context.

**Authors' response:** Thank you. We have further explained why we chose the DM method to bias correct the simulations of GCMs in this study. The added contents can be found in Page 5, Line 126-130, which also can be seen as follows:

"Considering that the distribution mapping (DM) method usually shows a comprehensive skill in bias correcting the mean,

standard deviation and various frequency-based indices and even correcting unobserved extreme values compared with other existing bias correction approaches like power transformation (PT), local intensity scaling (LOCI), linear scaling (LS), delta change (DC) and quantile mapping (QM) (Fang et al., 2015;Teutschbein and Seibert, 2012;Ji et al., 2020), the DM method was selected to correct GCM-simulated climate variables based on observations in this study."

**Comment #8:** L129. Please check the correctness of Eq. (2).

**Authors' response:** Thank you for your comment. There is indeed a small error in Eq. (2). The variable "$x_{sim,his}$" at the right hand side of the equation has been modified to "$x_{sim,fut}$".

**Comment #9:** L189. "is" <-> "was". The tense of this paper in the method part is a bit confusing. Please check the whole paper and ensure proper use of the tense.

**Authors' response:** Thank you. We have corrected this. In addition, we have checked the tense throughout the whole manuscript and have made the corresponding corrections, especially in the description of the used methods, including the sections of methods and results. The detailed revision can be found in the revised manuscript with changes tracked.

**Comment #10:** L338. In this paper, when investigating the changes of high flows and low flows, the 5-, 10- and 20-year return periods are adopted. Why not use the larger return periods such 50-year and 100-year return periods that are more useful information for assessment of extreme hydrological events?

**Authors' response:** To avoid introducing additional uncertainty through fitting observed and simulated 30-year high and low flows series with a probability distribution, the empirical cumulative distribution was directly used to calculate the values of high flows and low flows at different return periods in this study. Therefore, the largest return period calculated in this study is nearly 30 years, which obviously is smaller than 50 years and 100 years. This is why the high flows and

low flows at 50-year and 100-year return periods are not reported in this study.

**Comment #11:** L421. "account for approximately 54-60% on average" – Does this mean the proportion of the total uncertainty? Please make it clear.

**Authors' response:** Thank you. This is exactly the proportion of the total uncertainty. Therefore, we have modified the original content to "account for approximately 54-60% of the total uncertainty on average" in Page 24, Line 440.

**Referee #2:**

**Comment #1:** This manuscript only explained how to disaggregate monthly precipitation into daily precipitation considering the internal climate variability. How to downscale air temperature from monthly scale to daily scale should be also explained in the manuscript.

**Authors' response:** Thanks for your comments. In this study, we did not disaggregate monthly rainfall into daily rainfall, because we directly used the daily rainfall and temperature data of GCMs downloaded from https://data.cma.cn/ as described in Section 2. In fact, we spatially downscaled GCM simulated rainfall and temperature data in the historical and future periods rather than temporally downscaled the data, i.e. downscaling from the coarse GCM spatial resolution to the catchment resolution. This has been done using the distribution mapping (DM) method. The detailed downscaling methods are described in Section 3.1. In addition, this study indeed only considered the internal variability of rainfall through directly generating realizations of rainfall using the developed stochastic rainfall model SDRM-MCREM, the internal variability of temperature is not taken into consideration. Considering that the internal variability of rainfall is always large and comparable to or even greater than the recognized large uncertainty of GCMs (Hingray and Saïd, 2014; Giorgi, 2002), it should not be neglected in the analysis of different uncertainty sources in rainfall projections. Regarding the internal variability of temperature, several studies pointed out that it is usually very small compared with other uncertainty sources of temperature like GCM uncertainty and RCP uncertainty and therefore can be ignored without large consequences (Lafaysse et al., 2014; Hingray and Saïd, 2014; Fatichi et al., 2016). For that reason, we did not consider the internal variability of temperature in this study as an uncertainty source and this has also been explained in the discussion section (see Page 24, Line 435-439).

"Furthermore, the performances of GR4J model with the selected optimal parameter set in reproducing extreme flows including high flows and low flows are shown in Fig. 7. High flows are represented by annual maximum 1-day, 3-day and 5-day mean flow and low flows are represented by annual minimum 7-day, 30-day and 90-day mean flow. From Fig. 7, it can be seen that the exceedance probability distributions of high flows, particularly for the annual maximum 3-day and 5-day mean flow, are also very well reproduced by GR4J model with the optimum parameter set. The exceedance probability distributions of low flows are all slightly underestimated. Although the GR4J model slightly underestimates low flows, this underestimation can to some extent be eliminated as we use the relative change in results in the subsequent analyses."

[Figure]

**Figure 7:** Performances of GR4J model in reproducing the exceedance probabilities of high flows: (a) annual maximum 1-day flow, (b) annual maximum 3-day mean flow and (c) annual maximum 5-day mean flow and low flows: (d) annual minimum 7-day mean flow, (e) annual minimum 30-day mean flow and (f) annual minimum 90-day mean flow with the selected optimum parameter set in the historical period.

**Comment #3:** Figure 2: Replace 'dry spell' with 'Dry spell'. 'Modium' in the table should be 'Medium'.

**Authors' response:** Thank you. These have been modified accordingly, i.e. "dry spell" being replaced with "Dry spell" and "Modium" replaced with "Medium" in Figure 2, which can be seen as follows:

[Figure]

**Figure2:** Framework of SDRM-MCREM.

**Comment #4:** Figure 3 is not clear enough. Please enlarge the figures to improve the readability.

**Authors' response:** Thank you. We have redrawn Figure 3 to make it much clearer to read, which can be seen as follows:

[Figure]

**Figure 3:** Comparison of GCM simulations before and after bias correction with observations in the empirical cumulative distributions and monthly means of (a) daily rainfall, (b) daily maximum temperature, (c) daily minimum temperature and (d) daily mean temperature. The black line represents the observed values. The colorful lines represent raw GCM simulated values. The colorful dotted lines represent bias-corrected GCM simulated values.

**Comment #5:** Figure 4: In each sub-figure, there is a small plot box with three curves. Please denote the plot box in the figure caption.

**Authors' response:** Thank you very much. We have added a description for these small plot boxes to the caption of Figure 4, which is as follows:

[Figure]

**Figure 4:** Ensemble averages of GCM simulated monthly mean (a) rainfall, (b) daily maximum temperature, (c) daily minimum temperature and (d) daily mean temperature for the historical and two future periods under four RCPs after bias correction. The small graph in each sub-figure shows the ensemble averages of all GCMs and RCPs for the corresponding variable in the historical period and the two future periods, i.e. 2050s and 2080s.

**Comment #6:** Figure 6: Replace 'DRM-MCREM' with 'SDRM-MCREM'.

**Authors' response:** Sorry for the mistake. We have replaced "DRM-MCREM" with "SDRM-MCREM' in the caption of Figure 6.

**Comment #7:** Line 139: 'he features' should be 'the features'.

**Authors' response:** Thank you. We have modified "he features" to "the features".

[revised manuscript text omitted]

Third-order **Markov chain model** to generate time series of rainfall occurrence

**Extract rainfall events** and **determine rainfall duration** of each rainfall event

Stochastically **generate rainfall depth** for each rainfall event

**Determine the event class** of each rainfall event (Gao et al. 2019)

Stochastically **generate a rainfall type** and **simulate its corresponding rainfall pattern**

**Allocate rainfall duration and depth** according to the rainfall pattern for each rainfall event to obtain rainfall time series

Dry spell
Rainfall duration
Rainfall depth
Type A
Type C
Type D

[revised manuscript text omitted]